# Revisiting Sequential Information Bottleneck: New Implementation and Evaluation

**DOI:** 10.3390/e24081132

**Published:** 2022-08-16

**Authors:** Assaf Toledo, Elad Venezian, Noam Slonim

**Affiliations:** IBM Research AI, Haifa University Campus, Mount Carmel Haifa, Haifa 3498825, Israel

**Keywords:** clustering, information bottleneck, sequential algorithm

## Abstract

We introduce a modern, optimized, and publicly available implementation of the sequential Information Bottleneck clustering algorithm, which strikes a highly competitive balance between clustering quality and speed. We describe a set of optimizations that make the algorithm computation more efficient, particularly for the common case of sparse data representation. The results are substantiated by an extensive evaluation that compares the algorithm to commonly used alternatives, focusing on the practically important use case of text clustering. The evaluation covers a range of publicly available benchmark datasets and a set of clustering setups employing modern word and sentence embeddings obtained by state-of-the-art neural models. The results show that in spite of using the more basic Term-Frequency representation, the proposed implementation provides a highly attractive trade-off between quality and speed that outperforms the alternatives considered. This new release facilitates the use of the algorithm in real-world applications of text clustering.

## 1. Introduction

Unsupervised clustering of texts is a central problem in the domain of Natural Language Processing (NLP) [1,2,3], which has various applications in contemporary data analysis. For example, in the field of customer aid, clustering support tickets is helpful for identifying classes of user complaints and estimating their volume [4,5]. In research on public opinion, clustering data from social media such as Twitter or Reddit is useful for discovering active topics and learning about user engagement [6,7,8].

The Lloyd K-Means algorithm [9] is perhaps the most common choice for text clustering. It is also readily available in a modern, fast, and free implementation as part of the popular Scikit-Learn package [10]. The algorithm can be executed on top of a range of vector representations for the texts at hand, offering different trade-offs between clustering quality and speed.

The Term Frequency–Inverse Document Frequency (TF/IDF) [11] is a traditional method for textual data representation that was developed in the field of Information Retrieval. In this method, a text is first represented by a vector of term frequency (count) over a fixed vocabulary, and then, the value of each term is weighted by the inverse of its frequency in the overall corpus. In this weighting scheme, words that are common in the full corpus will be given a lesser weight compared to rare words when representing the text instance. This method is very fast to compute and process but suffers from the curse of dimensionality [12]. In addition, as a Bag-of-Words (BoW) method, it ignores word order.

More modern approaches that tackle the high dimensionality issue rely on Word Embeddings, such as Word2Vec [13,14] and GloVe [15]. In these cases, every word is assigned a fixed-size dense representation (embedding) trained by a neural network based on contextual information such as word–word co-occurrence statistics. With these methods, a common practice for obtaining the representation of a text instance is by averaging the embeddings of its words. These representations, however, still ignore word order, and while being relatively fast to generate and to work with, they disregard the sentential context in which words are meant to be interpreted.

With the surge of Deep Neural Networks [16] and advanced network architectures [17] and Language Models such as BERT [18], new records have been set in benchmarks of Natural Language Understanding [19,20]. Sentence-BERT (S-BERT) [21] is one of the latest advancements in creating models for general-purpose text vector representation that serve tasks such as semantic similarity, semantic search, and clustering. A common technique for creating these models is by adding a pooling layer to a BERT neural network. The pooling layer averages the contextualized word embeddings that BERT outputs and returns a single fixed-size vector for the whole text.

However, using an advanced language model typically comes at the expense of requiring more processing power and in many cases necessitates a GPU. The main concern in relying on vector representations from this kind of neural models is that in the age of Big Data, when the volume of texts is high and grows ever so quickly, it is critical for a clustering setup to be not only of high quality but also very efficient.

The sequential Information Bottleneck (sIB) [22] algorithm has shown strong results that outperform K-Means by a large margin on benchmark datasets such as [23]. However, sIB has never been available in a fast implementation that makes it comparable to K-Means for practical applications. Thus, despite being superior in lab testing, sIB was not commonly used in practice.

This paper introduces an efficient implementation of sIB that leverages a set of optimizations for obtaining a substantial improvement in speed compared to the original Matlab implementation. This is achieved while maintaining the same quality of clustering analysis provided by the original implementation.

The optimizations are focused on the case of sparse data representation, as sIB is usually running on top of sparse TF vectors, and it includes a mathematical derivation that computes the Jensen–Shannon divergence (JS) more efficiently on this type of representation. Our work builds on the analysis proposed in [24] and extends it to the case of a weighted JS, as this is the divergence used in sIB. In addition, we show an optimization that reduces the computational load by means of caching.

We present an evaluation scheme for assessing the quality and speed of the new sIB implementation and for comparing it against a set of competing clustering setups and over a range of benchmark datasets. Empirical results indicate that sIB is as fast as the fastest setups on most datasets, and in some cases, it is even the fastest. Quality-wise, sIB outperforms most competing setups by a large margin, and it maintains a small edge even over the best-performing setup of K-Means—when this algorithm leverages the advanced S-Bert representation.

In this manner, the new implementation strikes an attractive trade-off between quality and speed of text clustering, and it facilitates the use of sIB in real-world applications. The implementation is released as an open-source Python package under a permissive license (https://github.com/IBM/sib, available since 14 July 2020).

## 2. Algorithm Overview

In this section, we review the main properties of sIB, highlight key-differences between sIB and Lloyd K-Means, and put the focus on the main part of the algorithm that the new implementation optimizes.

### 2.1. Theoretical Foundation

sIB builds on the work of Tishby, Pereira, and Bialek [25] and views the clustering task as an Information Bottleneck (IB) problem. The algorithm first appeared in [22] and was explained in detail in [26]. Before formulating it, we present some assumptions and definitions.

sIB adopts the *Bag-of-Words* (BoW) approach with TF (count) vector representation for the texts to cluster. We use the following notation:*X*—the list of vectors to cluster;*Y*—the vocabulary used for representing the texts;p(X,Y)—the estimated joint distribution between *X* and *Y*;*K*—the number of clusters to produce;*T*—a partition of *X* into *K* clusters.

According to the IB method, given the joint distribution p(X,Y), we look for the partition *T*—in our case, a compressed representation of *X* into *K* clusters—that preserves as much information about *Y* as possible. Quoting from the original paper of sIB [22]:

“Intuitively, in this procedure the information contained in *X* about *Y* is ‘squeezed’ through a compact ‘bottleneck’ of clusters *T*, that is forced to represent the ‘relevant’ part in *X* with respect to *Y*”(p. 2)

Formally, the IB method is stated as:$$\min_{p(t|x)}I\left(X;T\right)-\beta I\left(T;Y\right)\;\;,$$
where I(X;T) and I(T;Y) are the mutual information (MI) between *X* and *T* and between *T* and *Y*, respectively. MI is defined as:$$I\left(A;B\right)=\sum_{a\in A,b\in B}p\left(a\right)\cdot p\left(b|a\right)\cdot log\frac{p(b|a)}{p(b)}\;\;.$$
The optimization is over the conditional probability p(t|x), and the non-negative parameter β is a Lagrange multiplier. The IB method [25] provides an exact optimal formal solution to this problem without any assumption about the origin of the joint distribution p(X,Y). In this analysis, the compactness of the representation *T* is determined by I(X;T), while the quality of *T* is measured by the fraction of the information that *T* and *X* capture about *Y*: I(T;Y)/I(X;Y). For a more detailed and technical discussion, see [22,25,26].

### 2.2. Divergence Function

As shown in [25], minimizing the IB functional defined above is obtained by using the Kullback–Leibler (KL) divergence [27] as the clustering divergence function between the conditional distributions p(y|x) and p(y|t):$$KL\left(p\left(y|x\right)∥p\left(y|t\right)\right)=\sum_{y}p\left(y|x\right)\frac{logp(y|x)}{p(y|t)}\;\;.$$

However, as shown in [26,28], in the hard-clustering setup—which is the focus of this work—to optimize the IB functional while merging *x* to *t*; one should use a weighted Jensen–Shannon divergence (JS) between p(y|x) and p(y|t). This divergence is defined based on the KL divergence with the additional features of being symmetric and always returning a finite value:$$JS\left(p\left(y|x\right)∥p\left(y|t\right)\right)=\pi_{1}\cdot KL\left(p\left(y|x\right)∥M\right)+\pi_{2}\cdot KL\left(p\left(y|t\right)∥M\right)\;\;,$$
where $M=\pi_{1}\cdot p\left(y|x\right)+\pi_{2}\cdot p\left(y|t\right)$, and $\pi_{1},\pi_{2}\in \left[0,1\right]$ are two weights such that $\pi_{1}+\pi_{2}=1$. For conciseness, we skip the definitions of $\pi_{1}$ and $\pi_{2}$ at this point and provide it in Section 3 where they are relevant for the formal analysis. The experimental setup in [22] confirms that the quality of the clustering analysis obtained when using JS as the clustering divergence function is superior to the clustering obtained when using KL. On the other hand, in terms of computational workload, JS is more demanding than KL as every computation of JS involves two computations of KL.

Let us compare this to K-Means. The traditional Lloyd K-Means algorithm is used with a geometrical distance function such as the Euclidean or cosine distance. This results in spatial clustering of the representations in a vector space. From a computational standpoint, the geometrical distances are lightweight and in some cases reduce to the computation of the dot product, which is fairly fast in comparison to the intensive log calculations as part of JS in sIB. Theoretically, this gives the traditional K-Means setup a substantial speed advantage over sIB.

The KL-Means algorithm [29,30,31] is a variant of the traditional algorithm with the distance function set to be the KL divergence. This effectively creates a version of K-Means that performs distributional clustering. It has been shown by [32] that in this setup, K-Means is algorithmically equivalent to the IB method where β→∞. In this sense, KL-Means is more similar to sIB than the traditional K-Means algorithm. sIB is still unique, however, in using JS divergence rather than KL divergence.

In the next section we provide the pseudo code for the algorithm and then move to another distinctive feature of sIB—namely, its sequential nature.

### 2.3. Pseudo-Code

The pseudo-code of the algorithm’s main loop is given in Algorithm 1, which is quoted from [22] with slight adjustments. The only modifications from the original pseudo-code are in the inner for-loop, where we explicitly mention the shuffle function and use *x* instead of $x_{j}$. The pseudo-code outlines the sequential workflow in which sIB works. In this code, recall that *K* is the number of clusters to generate, *n* is the number of (random) initializations, maxL is the maximal number of iterations per initialization, and ϵ is a lower bound threshold on the cluster updates for continuing to another iteration. In addition, shuffle is a function that randomizes the order of elements, *t* is used as a cluster identifier, *x* as a sample identifier, *c* is a counter of cluster changes during an iteration over *X* and *C* is a counter of iterations per initialization. Using several random initializations is a common practice with many clustering algorithms, as each initialization converges to a local maximum/minimum.

**Algorithm 1** Algorithm pseudo-code.


**Input:**


    |X| objects to be clustered
    Parameters: K,n,maxL,ϵ


**
Output:
**

    A partition *T* of *X* into *K* clusters


**
Main Loop:
**

     For i=1,…,n
            $T_{i}\leftarrow$ random partition of *X*.
            c←0,C←0,done=FALSE
            While not done
                 For *x* in *shuffle*(*X*)
                       draw *x* out of t(x)
                        $t^{new}\left(x\right)=\arg\min_{t^{\prime }}d_{F}\left(x,t^{\prime }\right)$
                        If $t^{new}\left(x\right)\neq t\left(x\right)$ then c←c+1
                        Merge *x* into $t^{new}\left(x\right)$
                  C←C+1
                  if C≥maxL or c≤ϵ·|X| then
                          done←TRUE
    $T\leftarrow \arg\max_{T_{i}}f\left(T_{i}\right)$


### 2.4. Sequential Clustering Algorithm

As shown in the pseudo-code, sIB is a sequential algorithm. This means that: (a) before selecting the new cluster for a sample, sIB withdraws that sample from its current cluster to prevent it from biasing the distance calculation toward keeping the sample in the same cluster, and (b) sIB updates the centroids while iterating over the samples and not only at the end of a full iteration over all samples.

Overall, while iterating over *X*, every sample is withdrawn from its cluster, the centroid of that cluster is updated, a new cluster is selected for that sample using the weighted JS divergence distance function, and then, the sample is added to the new cluster and the centroid of the new cluster is updated. In total, sIB performs 2·|X| centroid updates during a full iteration.

As discussed in detail in [33], this is a more powerful partition optimization method than the one employed by Lloyd K-Means, where there are no centroid updates while iterating over the samples during the assignment step. Lloyd K-Means performs only *K* centroid updates, which are all happening at the end of an iteration. Since K<<2|X| under normal circumstances, this gives Lloyd K-Means another substantial advantage in terms of computational workload.

### 2.5. Vector Representation

We distinguish between two vector representations: (a) for the texts to cluster and (b) for the centroids of clusters. Typically, the number of unique terms found in a specific text is much smaller than the vocabulary size. Therefore, it is more efficient to represent texts using sparse vector representations, both in terms of memory usage and processing time. In the sparse representation, it is sufficient to hold the list of IDs of vocabulary items found in the text and their frequency rather than an array of the size of the full vocabulary in which most of the values are zero. With regard to centroid vectors, as a centroid-based clustering algorithm, sIB constructs a centroid vector from the vectors of the samples that are associated with that cluster. Therefore, centroid vectors refer to a large part of the vocabulary and are encoded as regular non-sparse vectors.

### 2.6. Focus of This Work

This work focuses on the inner *for*-loop of the pseudo-code, which is the partition optimization part, and more specifically the computation of $t^{new}\left(x\right)$. We investigate it in the next section.

## 3. Methods

In this section, we present mathematical derivations and code optimizations that are at the center of the new implementation of sIB.

### 3.1. Computation of $t^{new}\left(x\right)$ and Associated Intuition

Recall that finding the new cluster assignment for *x* relies on computing
(1)$$t^{new}\left(x\right)=\underset{t}{arg min}d_{F}\left(x,t\right)\;\;,$$
where $d_{F}$ is given by
(2)$$d_{F}\left(x,t\right)=\left(p\left(x\right)+p\left(t\right)\right)\cdot JS\left(p\left(y|x\right),p\left(y|t\right)\right)\;\;,$$
and JS is a weighted *Jensen–Shannon divergence* defined with weights $\pi_{1}$ and $\pi_{2}$:(3)$$\pi_{1}=\frac{p(x)}{p(x)+p(t)},\;\;\;\;\pi_{2}=\frac{p(t)}{p(x)+p(t)}\;\;.$$

Intuitively, when selecting the new cluster assignment for *x*, we examine the distribution over the vocabulary induced by *x* (p(y|x)) and compare it to the distribution over the vocabulary induced by each cluster’s centroid (p(y|t)) using the weighted JS divergence multiplied by p(x)+p(t). The cluster $t^{new}$ is selected as the cluster for which this multiplication is minimized.

In what follows, we use the following notation:$\hat{x}$ = p(y|x)—the TF vector representing the sample *x* normalized by the $L^{1}$-norm. Let $u\in R^{n}$, the $L^{1}$-norm ${\left|u\right|}_{1}$ of *u* is defined by: ${\left|u\right|}_{1}=\sum_{i=1}^{n}\left|u_{i}\right|$;$\hat{t}$ = p(y|t)—the vector representing the centroid of cluster *t*, normalized by $L^{1}$;*m* = $\pi_{1}\cdot \hat{x}+\pi_{2}\cdot \hat{t}$—the average of $\hat{x}$ and $\hat{t}$ weighted by $\pi_{1}$ and $\pi_{2}$, respectively.

Using these notations:(4)$$JS\left(\hat{x},\hat{t}\right)=\pi_{1}\cdot KL\left(\hat{x}∥m\right)+\pi_{2}\cdot KL\left(\hat{t}∥m\right)\;\;,$$
where KL is the *Kullback–Leibler divergence* [27] defined as:(5)$$KL\left(u∥v\right)=\sum_{i}u\left[i\right]\cdot log\left(\frac{u[i]}{v[i]}\right)\;\;.$$

Following the analysis in [24], simple algebra gives the form in (Equation 6):(6)$$JS\left(\hat{x},\hat{t}\right)=H\left(m\right)-\pi_{1}\cdot H\left(\hat{x}\right)-\pi_{2}\cdot H\left(\hat{t}\right)\;\;,$$
where *H* is Shannon’s entropy function: $H\left(u\right)=-\sum_{i}u\left[i\right]\cdot log\left(u\left[i\right]\right)$.

Since $\left(p\left(x\right)+p\left(t\right)\right)\cdot \pi_{1}=p\left(x\right)$ and $\left(p\left(x\right)+p\left(t\right)\right)\cdot \pi_{2}=p\left(t\right)$:(7)$$\begin{array}{cc} d_{F}\left(x,t\right) & =\left(p\left(x\right)+p\left(t\right)\right)\cdot H\left(m\right)-p\left(x\right)\cdot H\left(\hat{x}\right)-p\left(t\right)\cdot H\left(\hat{t}\right)\;\;. \end{array}$$
Because $p\left(x\right)\cdot H\left(\hat{x}\right)$ is a constant with respect to *t*, we get:(8)$$\underset{t}{arg min}\left(d_{F}\left(x,t\right)\right)=\underset{t}{arg min}\left(\left(p\left(x\right)+p\left(t\right)\right)\cdot H\left(m\right)-p\left(t\right)\cdot H\left(\hat{t}\right)\right)\;\;.$$

To obtain some insight into how sIB selects the cluster *t* for a sample *x*, we examine two pairs of components in Equation (Equation 8)—(a) H(m) and $H(\hat{t})$; and (b) (p(x)+p(t)) and p(t). Starting with (a), since *m* is a weighted average of $\hat{x}$ and $\hat{t}$, a better fit of *x* to *t* implies lower discrepancy between *m* and $\hat{t}$, which in turn results in a smaller difference between H(m) and $H(\hat{t})$. Thus, the preference is for selecting a cluster *t* that represents a good fit for *x*. Moving to (b), as the cluster *t* increases, the relative difference between p(x)+p(t) and p(t) decreases. Therefore, the components in (b) can be seen as balancing factors for the selection of *t* by taking into account the size of the cluster and giving preference to larger clusters. Typically, these two parts compete, since as *t* becomes larger, it often also becomes less distinctive; hence, it is harder for it to provide a good fit for *x*.

### 3.2. Optimization for Sparse Vector Representation

In this section, we show a computation of $t^{new}\left(x\right)$ that is optimized for sparse vector representation. Let $x_{ind}$ be the indices of non-zero values in *x*. As explained in Section 2.5, a sparse representation is the natural choice for TF vectors since typically $|x_{ind}\left|<<|Y\right|$.

We evaluate Equation (Equation 6) as:(9)$$JS\left(\hat{x},\hat{t}\right)=\sum_{i\in x_{ind}}R_{i,\hat{x},\hat{t}}+\sum_{i\notin x_{ind}}R_{i,\hat{x},\hat{t}}\;\;,$$
where $R_{i,\hat{x},\hat{t}}$ is defined as:(10)$$R_{i,\hat{x},\hat{t}}:=\pi_{1}\cdot {\hat{x}}_{i}\cdot log\left({\hat{x}}_{i}\right)+\pi_{2}\cdot {\hat{t}}_{i}\cdot log\left({\hat{t}}_{i}\right)-\left(\pi_{1}\cdot {\hat{x}}_{i}+\pi_{2}\cdot {\hat{t}}_{i}\right)\cdot log\left(\pi_{1}\cdot {\hat{x}}_{i}+\pi_{2}\cdot {\hat{t}}_{i}\right)\;\;.$$

Since the computation of $R_{i,\hat{x},\hat{t}}$ involves a constant number of operations, the first component in (Equation 9) has a computational complexity of $O(|x_{ind}|$). Let us now evaluate the second component and show that it has the same complexity.

By definition, $\forall i\notin x_{ind}\;\;{\hat{x}}_{i}=0$. Consequently,
(11)$$\sum_{i\notin x_{ind}}R_{i,\hat{x},\hat{t}}=\sum_{i\notin x_{ind}}\left[\pi_{2}\cdot {\hat{t}}_{i}\cdot log\left({\hat{t}}_{i}\right)-\pi_{2}\cdot {\hat{t}}_{i}\cdot log\left(\pi_{2}\cdot {\hat{t}}_{i}\right)\right]\;\;.$$
Since $log\left(\pi_{2}\cdot {\hat{t}}_{i}\right)=log\left(\pi_{2}\right)+log\left({\hat{t}}_{i}\right)$,
(12)$$\sum_{i\notin x_{ind}}R_{i,\hat{x},\hat{t}}=\sum_{i\notin x_{ind}}\left[\pi_{2}\cdot {\hat{t}}_{i}\cdot log\left({\hat{t}}_{i}\right)-\pi_{2}\cdot {\hat{t}}_{i}\cdot \left(log\left(\pi_{2}\right)+log\left({\hat{t}}_{i}\right)\right)\right]\;\;.$$
With simple algebra, we obtain:(13)$$\sum_{i\notin x_{ind}}R_{i,\hat{x},\hat{t}}=-\pi_{2}\cdot log\left(\pi_{2}\right)\cdot \sum_{i\notin x_{ind}}{\hat{t}}_{i}\;\;.$$
Since $\hat{t}$ is normalized by $L^{1}$-norm, $\sum_{i}{\hat{t}}_{i}=1$. Therefore:(14)$$\sum_{i\notin x_{ind}}R_{i,\hat{x},\hat{t}}=-\pi_{2}\cdot log\left(\pi_{2}\right)\cdot \left(1-\sum_{i\in x_{ind}}{\hat{t}}_{i}\right)\;\;.$$
This means that the second component of (Equation 9) also has a computational complexity of $O(|x_{ind}|)$, which shows that (Equation 6) and consequently (Equation 2) have the same computational complexity of $O(|x_{ind}|)$. In the non-sparse case, the computational complexity is O(|Y|), which is significantly higher.

With respect to (Equation 1), since there are *K* centroids to select from, the overall complexity of computing $t^{new}\left(x\right)$ is $O(K\cdot |x_{ind}|)$ in the sparse case and O(K·|Y|) in the non-sparse case.

### 3.3. Caching Log Computations

The most time-consuming operation in the computation of a new cluster for a sample is the log function. In this section, we show a way to reduce the number of log computations via caching. Let us recall Equation (Equation 8), which is repeated below:$$\underset{t}{arg min}\left(d_{F}\left(x,t\right)\right)=\underset{t}{arg min}\left(\left(p\left(x\right)+p\left(t\right)\right)\cdot H\left(m\right)-p\left(t\right)\cdot H\left(\hat{t}\right)\right)\;\;.$$

We observe that $H(\hat{t})$ is independent of the sample *x* for which we calculate the new cluster. Thus, we can cache this computation and reuse it when we iterate over the samples. When a sample is drawn out of a cluster or merged into a cluster, we update only the entries in the cache that refer to the clusters that have been updated. The gain can be summarized as follows. Given a sample *x* for which we compute a new cluster, instead of computing the entropy over all centroids, we compute it only for two centroids—the centroid of the cluster from where *x* is drawn out and the one to which *x* is merged into.

### 3.4. Implementation

The new implementation of sIB is based on the optimizations presented above. See Appendix A for information about the source-code availability and its Python packaging.

## 4. Experimental Setup

We evaluate the quality and speed of the new implementation of sIB against the robust open source implementation of Lloyd K-Means [9] from Scikit-Learn [10]. We use a set of five datasets that are common in text classification and text clustering benchmarks and measure the clustering quality by standard clustering metrics. We use multiple setups for K-Meams, each on top of a different vector representation type.

### 4.1. Materials

In order to cover various use cases, we employ datasets of different source, size, text length and number of classes. The datasets are described below, and statistical information is summarized in Table 1. All datasets are publicly available online at the locations specified in the Data Availability Statement.

*BBC News* [34] consists of 2225 articles from the BBC news website. The articles are from 2004–2005 and cover stories in five topical areas: *business, entertainment*, *politics*, *sport*, and *tech*.*20 News Groups* [23] consists of 18,846 emails sent through 20 news groups. The topics are diverse and cover *tech, religion,* and *politics*, among others.*AG News* [35] consists of 127,600 pairs of titles and snippets of news articles from the AG corpus, covering four topical areas: *World*, *Sports*, *Business*, and *Sci/Tech*. The title and snippet of each article are concatenated when the data is clustered.*DBPedia* [35] consists of 630,000 pairs of titles and abstracts of documents from 14 non-overlapping ontology classes such as *Artist*, *Film*, and *Company*. The title and abstract are concatenated when the data are clustered.*Yahoo! Answers* [35] consists of 1,460,000 triplets of question title, question content, and best answer from the Yahoo! Answers Comprehensive Questions and Answers version 1.0 dataset. The data covers the 10 largest topical categories, such as *Society & Culture*, *Computers & Internet*, and *Health*. The question title, content, and best answer are concatenated when the data are clustered.

### 4.2. Clustering Metrics

We use five metrics to evaluate the clustering quality: (a) Adjusted Mutual-Information (AMI): the mutual-information corrected for chance [36,37], (b) Adjusted Rand-Index (ARI): the rand index corrected for chance [36], (c) V-Measure: the harmonic mean between homogeneity and completeness [38], (d) Micro-F1: the micro average of F1 scores over all classes in the dataset, and (e) Macro-F1: the macro average of F1 scores over all classes in the dataset. All metrics are calculated against the ground-truth labels of each dataset.

### 4.3. Clustering Setups

sIB runs on top of sparse TF representations. The encoding is done using a vocabulary of the 10,000 most common words in each dataset after stop-words filtering. We use the Scikit-Learn [10] TF encoder. The algorithm runs with 10 random partitions of equally sized clusters in parallel. Each initialization is optimized by up to 15 iterations or until the number of samples changing cluster is less than 2% (all are default values). This generates 10 partitions of the data, and the algorithm returns the partition that maximizes *I*(*T*;*Y*)/*I*(*X*;*Y*) as explained in Section 2.1. The sIB version is 0.1.8, and the Scikit-Learn version is 1.1.1.

K-Means runs on top of several representations: TF, TF/IDF [11], GloVe [15] mean vectors and Sentence-Bert (S-Bert) [21]. The TF is the same as used for sIB (described above). The TF/IDF representation is generated using Scikit-Learn [10] TF/IDF encoder with the same settings as TF and is also sparse. For GloVe, each text is represented by averaging the embeddings of its words after punctuation and stop-words filtering. We use the *glove-840b-300d* pre-trained model, which was trained on 840 billion tokens and produces 300-dimensional dense vectors. For S-Bert, we employ the pre-trained model *all-MiniLM-L6-v2* which aims to provide a fine balance between quality and processing time. This model was trained on 1 billion sentence pairs and produces 384-dimensional dense vectors.

We use the Scikit-Learn [10] Lloyd K-Means implementation in its default settings. The algorithm runs with 10 random centroid initializations obtained by K-Means++ [39] in parallel, yielding 10 partitions of the data. Each initialization is optimized by up to 300 iterations or until the centroids movement between iterations, as measured by Frobenius norm, is less than $10^{-4}$. The algorithm returns the partition that minimizes the sum of distances between each sample and the centroid of its cluster. All are default values. The distance function used in this version of K-Means is the squared Euclidean distance [40]. Minimizing the squared distance is equivalent to minimizing the Euclidean distance since squaring is a monotonic function of non-negative values. Oftentimes, the squared distance is preferred because it is faster to compute.

### 4.4. Robustness Considerations

Since both sIB and K-Means rely on random initialization, every run of these algorithms converges to a different local minimum and yields a different clustering result. For robustness, we run every setup described above 10 times and apply the metrics described in Section 4.2 to every such run. We obtain 10 scores for each metric for a given setup and report only the average score per metric. We also use the distribution of the metric scores for calculating confidence intervals.

### 4.5. Hardware

The hardware used is a MacBook Pro 2019 with an 8-Core Intel Core i9 running at 2.3 Ghz. Hyper-threads: 16. Memory is 64 GB 2667 MHz DDR4. This simulates a local run by a data scientist.

### 4.6. Code

The evaluation code is available on the sIB open source repository and can be extended and tweaked to cover more algorithms, representations, and settings.

## 5. Results

The results are detailed in Table 2. In terms of clustering quality, the metrics indicate that sIB has the edge over the setup of K-Mean on top of S-Bert on the 20 News Groups and AG News datasets. On the BBC News dataset they are even, and then the trend reverses and K-Means on top of S-Bert takes the lead by a relatively small margin on the DBPedia and Yahoo! Answers datasets. Overall, these two setups are roughly on par with a slight edge to sIB. The other K-Means setups are trailing behind by a large margin, with the GloVe setup being better than the TF/IDF setup, and the TF setup being the weakest. Figure 1 illustrates the results on the AMI and ARI metrics. We include charts also for the Micro-F1, Macro-F1 and V-Measure in Appendix B. As explained in Section 4.4, the reported result of every metric is the average of 10 runs of each setup. Error bars in the figures indicate the 95% confidence interval obtained by bias-corrected and accelerated (BCa) bootstrapping of the 10 results per metric.

As for run-time measurements, we can see in Table 2 that sIB is as fast as the quickest K-Means setups (TF and TF/IDF) on the datasets of 20 News Groups, AG News, BBC News and DBPedia, and it is the fastest setup on the Yahoo! Answers dataset. sIB is also faster than the setup of K-Means on top of GloVe by a noticeable margin.

The setup of K-Means on top of S-Bert, which is the only setup that is competitive with sIB quality-wise, is substantially slower due to the neural vectorization on CPU. On average, this setup is 200 times slower than sIB. More generally, the S-Bert model is more power demanding than any other representation type evaluated here, and for practical use cases, especially on large datasets such as DBPedia and Yahoo! Answers, it is likely to necessitate a GPU or even more than one. A chart of the total run-time measurements is included in Appendix B.

### Discussion

The results emphasize the premise of the sIB implementation proposed in this work: delivering a clustering analysis that is as good as can be obtained by a state-of-the-art neural model while being far less demanding in terms of run-time. In this way, sIB offers a more attractive trade-off between quality and speed than the rest of the setups evaluated here.

Looking at the run-time measures in absolute terms, sIB is able to cluster the 630,000 texts of DBPedia in about 1 minute and the 1,460,000 texts of Yahoo! Answers in about 3.5 min using standard CPU hardware. Both are very practical and workable run-times for real-world applications.

A question can be raised as to how sIB can match or even improve on the K-Means run-time. Given that sIB is a more demanding algorithm in terms of computational workload, it would have been expected to show inferior run-time measurements compared to the more lightweight K-Means. We look into this in Appendix C and provide a hypothesis to this phenomenon. Note that we ignore here the discrepancy between sIB and K-Means with respect to the default maximal number of partition optimization iterations (15 for sIB, 300 for K-Means). This is because internal testing with fewer iterations for K-Means proved ineffective for reducing the algorithm run-time. We assume that this is because the algorithm declared convergence (to a local minimum) long before the iteration limit is reached.

## 6. Conclusions

The sIB algorithm was introduces more than 15 years before the rise of the language models revolution in NLP. Although sIB uses simple TF representations, it utilizes a powerful probabilistic framework and a robust optimization method. This work is the first to offer a highly efficient implementation of the algorithm and also to evaluate it on contemporary benchmark datasets against competing, more popular, clustering setups.

Empirical results indicate that sIB creates a high-quality clustering analysis, which is comparable to the level of analysis obtained when using representations from a state-of-the-art language model. Speed-wise, the results show that the new implementation enables users to easily run sIB on a standard CPU hardware, and that it is far less demanding than a neural solution. In this manner, sIB offers an attractive trade-off between quality and speed, outperforming the rest of the setups considered in this work.

In the future, we plan to look into new ways to reduce sIB’s run-time further by creating “lossy” modes of the algorithm. In such modes, rather then iterating over all samples per iteration, the algorithm can allow certain samples to be skipped based on information from previous iterations. For example, if a sample remains in the same cluster for several consecutive iterations, or if it fits much better in one cluster compared to the others, it can be considered as locked-in in its current cluster. In this manner, one can further reduce the algorithm run-time and offer more control in tuning the desired trade-off between quality and speed, allowing sIB to fit an even broader set of use-cases and reach a wider audience.

The new implementation of sIB is released as open-source under a permissive license, and it can be integrated as part of a more complex pipeline of natural language processing in research projects as well as in real-world applications. We hope that practitioners of text clustering and researchers interested in the IB line of study will find this work and the released code valuable.

## Figures and Tables

**Figure 1 entropy-24-01132-f001:**
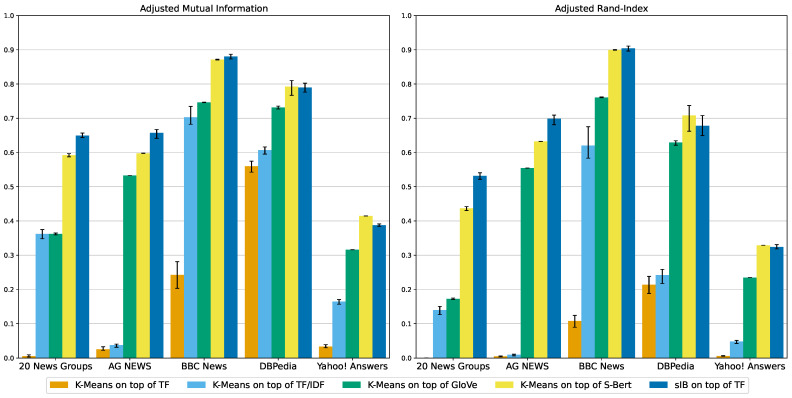
Adjusted Mutual-Information (**left**) and Adjusted Rand-Index (**right**) scores for the clustering setups over all benchmark datasets. The scores are means of 10 samples per metric. Error bars indicate the 95% confidence interval obtained by bias-corrected and accelerated bootstrapping.

**Table 1 entropy-24-01132-t001:** Benchmark datasets for evaluation. The column *#Texts* indicates the number of texts in the dataset. The column *#Words* shows the average text length in terms of word count in the dataset, and *#Classes* shows the number of classes in the dataset.

Dataset	#Texts	#Words	#Classes
BBC News	2225	390	5
20 News Groups	18,846	284	20
AG NEWS	127,600	38	4
DBPedia	630,000	46	14
Yahoo! Answers	1,460,000	92	10

**Table 2 entropy-24-01132-t002:** Assessment of clustering quality using the metrics: AMI, ARI, V-Measure *(VM*), Micro-F1 (*Mic-F1*) and Macro-F1 (*Mac-F1*), and of clustering speed based on measurements of the vectorization time (*Vector*), clustering time (*Cluster*), and their sum (*Total*).

Dataset	Algorithm	Embed	AMI	ARI	VM	Mic-F1	Mac-F1	Vector	Cluster	Total
20 News Groups	K-Means	TF	0.01	0.00	0.01	0.06	0.01	**00:03**	**00:07**	**00:10**
	K-Means	TF/IDF	0.36	0.14	0.36	0.35	0.32	**00:03**	00:10	00:14
	K-Means	GloVe	0.36	0.17	0.36	0.34	0.31	00:19	00:09	00:28
	K-Means	S-Bert	0.59	0.44	0.59	0.61	0.58	22:27	00:08	22:35
	sIB	TF	**0.65**	**0.53**	**0.65**	**0.66**	**0.61**	**00:03**	00:11	00:14
AG NEWS	K-Means	TF	0.03	0.00	0.03	0.29	0.20	**00:03**	**00:02**	**00:05**
	K-Means	TF/IDF	0.04	0.01	0.04	0.31	0.24	**00:03**	00:03	00:06
	K-Means	GloVe	0.53	0.55	0.53	0.80	0.80	00:19	00:07	00:26
	K-Means	S-Bert	0.60	0.63	0.60	0.84	0.84	38:18	00:11	38:29
	sIB	TF	**0.66**	**0.70**	**0.66**	**0.87**	**0.87**	**00:03**	00:03	00:06
BBC News	K-Means	TF	0.24	0.11	0.24	0.41	0.32	00:01	**00:00**	**00:01**
	K-Means	TF/IDF	0.70	0.62	0.70	0.83	0.83	**00:00**	**00:00**	**00:01**
	K-Means	GloVe	0.75	0.76	0.75	0.90	0.90	00:05	**00:00**	00:06
	K-Means	S-Bert	0.87	**0.90**	0.87	**0.96**	**0.96**	02:55	**00:00**	02:56
	sIB	TF	**0.88**	**0.90**	**0.88**	**0.96**	**0.96**	00:01	00:01	**00:01**
DBPedia	K-Means	TF	0.56	0.21	0.56	0.50	0.47	**00:20**	**00:43**	**01:04**
	K-Means	TF/IDF	0.61	0.24	0.61	0.56	0.55	**00:20**	00:45	01:06
	K-Means	GloVe	0.73	0.63	0.73	0.76	0.72	01:28	02:08	03:37
	K-Means	S-Bert	**0.79**	**0.71**	**0.79**	**0.82**	**0.79**	03:38:31	02:00	03:40:31
	sIB	TF	**0.79**	0.68	**0.79**	0.78	0.74	**00:20**	00:44	01:05
Yahoo! Answers	K-Means	TF	0.03	0.01	0.03	0.15	0.08	**01:15**	04:22	05:37
	K-Means	TF/IDF	0.16	0.05	0.16	0.29	0.25	01:16	03:44	05:01
	K-Means	GloVe	0.32	0.23	0.32	0.49	0.44	06:21	06:42	13:03
	K-Means	S-Bert	**0.41**	**0.33**	**0.41**	**0.59**	**0.56**	16:20:10	06:18	16:26:28
	sIB	TF	0.39	0.32	0.39	0.57	0.54	**01:15**	**02:20**	**03:35**

## Data Availability

Publicly available datasets were analyzed in this study. These datasets were accessed on 30 June 2022 at: 20 News Groups: https://scikit-learn.org/stable/datasets/real_world.html#the-20-newsgroups-text-dataset; AG News: https://huggingface.co/datasets/ag_news; BBC News: https://huggingface.co/datasets/SetFit/bbc-news; DBPedia: https://huggingface.co/datasets/dbpedia_14; Yahoo! Answers: https://huggingface.co/datasets/yahoo_answers_topics.

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
