# Peer review of "Revisiting Sequential Information Bottleneck: New Implementation and Evaluation"

_entropy, 2022, doi:10.3390/e24081132_

Round 1
Reviewer 1 Report
This paper revisits the sequential Information Bottleneck clustering algorithm, providing additional optimisation and implementation guidelines. The results show that this algorithm is comparable in effectiveness to k-means using sBert without incurring the same computational/efficiency cost. The work is adequately supported in terms of scientific and experimental justification.
The main issue I can identify is that the study lacks originality. It does not improve or suggest a novel algorithm or novel parameterisation of an existing algorithm, but it provides an optimised implementation. The implementation optimisation itself is underwhelming, mainly via using sparse representations (which I suppose is the norm for document clustering) and by caching log computations - as opposed to e.g. via a novel use of a data structure, or offering a parallel version of the algorithm, etc. It is due to this that I am not sure how interesting this article would be to readers.
In terms of language, presentation and justification of the choices made, this article is well written.
Minor points:
> l.106-107 - I am not sure that the choice of euclidean distance in k-means is an "adhoc" one. It follows from the definition of a centroid. Cosine similarity is directly related to euclidean distance, so it's not inherently different
> l.145 - remove "As for centroid vectors"
> Since the experiments are on relatively small and openly available collections, it would be very useful if you could also provide the evaluation source code, so that readers can reproduce table 2.
Author Response
Thank you for the review. Please find our response in the attached file.

Reviewer 2 Report
The authors present an improved implementation of the sequential Information Bottleneck (sIB) algorithm. To this end, they describe the necessary algorithmic steps to achieve a more efficient algorithmic implementation with focus on sparse data representations. The authors ensure that the common reader can follow each modification step. In addition, the developed algorithm is made publicly available as a Python package.
The performance of the developed algorithmic implementation is investigated for the case of text clustering and compared to other schemes from literature. To this end, five different data sets common in text classification and text clustering are used. It is demonstrated that the proposed sIB implementation outperforms or at least performs similar to K-means on to of s-Bert, but consuming less run-time on CPUs. Thus, an attractive trade-off between quality and speed is achieves.
The authors discuss the benefits compared to the K-Means algorithm. However, the applied distance measure for the K-Means remains unclear for the reviewer. Notice, that the “KL-Means” replaces the common Euclidean distance by KL-divergence [A] and the equivalence to the IB method for \beta=0 has been demonstrated [B].
It would be good to introduce the IB problem in section 2.1 formally.
Detailed comments:
· Line 96: The notation for mutual information “I<T;Y>” is rather uncommon. Why not using I(T;Y)”. Furthermore, please elaborate on the quality measure “I<T;Y>/I<X;Y>”.
· Line 105:
· Line 164: L_1 has not been defined
· Line 169: use italic symbol “H” in “where H is Shannon’s entropy function”
· Line 186: In order to avoid confusion with mutual information I(T;Y), it could be an option to use another letter for (10).
· Line 189: The variable I_i has been used to define I_i,\hat{x},\hat{t} for i\notin x_ind. It is recommended to introduce I_i correspondingly.
Additional literature
[A] A. Zhang and B. M. Kurkoski, “Low-complexity quantization of discrete memoryless channels,” ISITA 2016
[B] B.M. Kurkoski, “On the Relationship Between the KL Means Algorithm and the Information Bottleneck Method”
Author Response

(The authors gave the same response as above.)
